# Effectiveness of Expressive Writing in Kidney Transplanted Patients: A Randomized Controlled Trial Study

**DOI:** 10.3390/healthcare10081559

**Published:** 2022-08-17

**Authors:** Laura Pierro, Giulia Servidei, Renzo Pretagostini, Davide Stabile, Francesco Nudo, Silvia Lai, Paola Aceto, Luca Poli, Erika Fazzari, Carlo Lai

**Affiliations:** 1Department of Dynamic and Clinical Psychology and Health Studies, Via degli Apuli 1, Sapienza University, 00185 Rome, Italy; 2Department of General Surgery “Paride Stefanini”, Viale del Policlinico 155, Sapienza University, 00161 Rome, Italy; 3Department of Clinical Medicine, Viale dell’Università 37, Sapienza University, 00185 Rome, Italy; 4Department of Anaesthesiology and Intensive Care Medicine, Via della Pineta Sacchetti 217, Fondazione Policlinico Universitario A. Gemelli IRCCS, 00168 Rome, Italy

**Keywords:** psychological intervention, graft rejection, renal transplantation, randomized control trial, health psychology

## Abstract

The present study aimed to assess the effectiveness of an expressive writing (EW) intervention on psychological and physiological variables after kidney transplant. The final sample of 26 were randomly assigned to an expressive writing group (EWG) and control group (CG). Outcomes were focused on depression, anxiety, alexithymia, empathy, resilience, locus of control, creatinine, CDK-EPI, and azotemia. Depressive symptoms and alexithymia levels decreased in the EWG, with better adherence. Resilience declined over time in both groups. The EWG showed a significantly higher CDK-EPI, indicating better renal functioning. EW seems an effective intervention to improve the psychological health of transplanted patients, with a possible effect on renal functioning. These findings open the possibility of planning brief psychological interventions aimed at processing emotional involvement, in order to increase adherence, the acceptance of the organ, and savings in healthcare costs.

## 1. Introduction

Kidney transplantation is a replacement procedure for chronic kidney disease that is intended to improve the patient’s quality of life. However, the experience may be a strongly emotional one, accompanied by difficulty in accepting the new organ, lifestyle changes, the side-effects associated with the procedure, anxiety, depression, the redefinition of one’s body, and a change in social and family roles [1,2,3,4,5]. Hence, the associated stress may result in an adverse post-operative course, particularly in terms of compliance to the follow-up process, medical prescriptions, and the consequential risk of rejection [6].

Adherence is described as the correct management, conforming to the prescribed anti-rejection drug (immunosuppressive) therapy, maintaining regularity in visiting the pre-established check-ups at the transplant center, and prompt reporting of any complications [4,7]. In kidney transplanted patients, adherence to immunosuppressive therapy is of importance, to reduce the risk of rejection [8]. Several studies have highlighted the role of adherence to treatment on the postoperative course in different transplantation procedures [9], with special emphasis on the association of lower adherence levels with an increase in healthcare costs [10].

Younger age, depression and anxiety, using avoidance as an adaptation strategy, and the inability to communicate emotions and emotional dysregulation represent some of the factors predisposing to poor medication adherence in transplant patients [3,11,12]. 

Expressive writing seems to be a suitable intervention in transplanted patients, as previously shown by Possemato et al. [13], with its effects on post-traumatic stress symptoms and quality of life. This technique has also been proven effective in patients with chronic diseases and cancer, to reduce and manage psychophysical symptoms [14,15]. The externalization and processing of stressful events related to the disease probably encourage the ability to express emotions, thereby reducing the associated negative thoughts and feelings [16]. 

The present study aimed to evaluate the effect of expressive writing on psychological variables and on the biological markers of renal function after transplant. It was hypothesized that the expressive writing intervention would be effective in improving psychological evaluation, enabling better adherence, lowering the risk of organ rejection, and decreasing health expenditure.

## 2. Materials and Methods

### 2.1. Study Design

Parallel-group randomized trials were applied. The study complied with the guidelines outlined under the Consolidation Standards of Reporting Trials (Extension of the CONSORT 2010 Statement) checklist. The study involved two parallel groups: experimental and control. The samples were composed using a double-blind randomized controlled trial (RCT). The trial was registered on https://www.clinicaltrials.gov/ Reg. no. NCT04486417 with receipt on 12 November 2019 (first posted 29 October 2019).

The study was approved by the Ethics Committee of the Department of Dynamic and Clinical Psychology and Health Studies, Sapienza University of Rome (approval number 23/2017). Patients who agreed to participate signed a written consent form obtained for experimentation with human subjects, in accordance with the 1964 Declaration of Helsinki.

### 2.2. Participants

Fifty-seven renal-transplant patients at the Kidney Transplant Center, Policlinico Umberto I, “Sapienza” University of Rome, were contacted between March 2018 and March 2019. Participants aged above 18 years of age with sufficient ability to read, comprehend, and write were considered for the study. Those with severe psychopathological conditions already encountered during the pre-transplant psychiatric evaluation and those with cognitive impairment were excluded. The first contact was with the clinicians of the hospital, who explained to the consulted patients the aim of the research and the clinical aspects of the study, and then established the inclusion and exclusion criteria.

Out of the 57 patients contacted, 35 were recruited initially, of which 2 had an acute rejection before the randomization, and 9 others dropped out, leaving a final sample of 26. The sample size was based on an a priori power analysis, which considered the effect reported in a previous study on similar outcomes, such us depression [17], with a significance level of *α* = 0.05, *β* = 0.02 and power (1-*β*) = 80%. This suggested a sample size of 10 participants (5 per group), therefore the number of participants tested in the present study can be considered adequate. The participants were blindly randomly assigned to one of the two groups by an independent researcher. All participants were randomized into two groups; the random allocation of the participants was performed in a double-blind manner, the reference researcher generated the random allocation sequence, asking the participants to extract from a box a closed sheet, in which the kind of group was specified. The sheets were filled by a clinician and placed in the box. Neither the researcher nor the participant knew a priori the contents of the extracted sheets. Participants were not informed about the treatment tasks. The size of the randomization block was established as reaching 5 consecutive extractions with the same group; at which point, the group was eliminated from the box, and the extraction continued with only one sheet. The two groups were the expressive writing group (EWG) and the control group (CG).

### 2.3. The Writing Interventions

The expressive writing process consisted of writing on 3 different topics, 20-min every day for 3 consecutive days. On day 1, the EWG participants were asked to write about their deepest emotions, thoughts, and concerns related to the disease and the transplant itself, followed by an account of the most difficult experience of their life and its association with the onset of the kidney disease on day 2, and finally about their future expectations after the operation on day 3. Whereas the CG participants were asked to write about an assigned neutral theme (describe an object in the room), in the most objective way possible, without mentioning emotions or thoughts related to it. The trial was stopped at the end of the 20 min.

### 2.4. Procedure

The study comprised 3 different phases: 

(1) Pre-operative (T0): corresponding to the hospitalization day, just 2 h before the transplantation.

(2) Psychological intervention (IP): Five days after the transplant, all participants were asked to choose one of the two envelopes, containing the expressive or neutral writing tasks, provided by the researcher, to randomly allocate the writing task. The researcher remained available for clarifications, after which the patient was left alone for 20 min to complete the writing task. The same procedure was applied for the next 2 days for both groups.

(3) Post-operative phase consisting of 2 sub-phases: (a) T1-phase, the day of discharge; (b) T2-phase, the 3rd post-operative month.

### 2.5. Outcome Measures

All participants were invited to complete the following questionnaires at times T0, T1, and T2:

Beck Depression Inventory-II (BDI-II) [18]; State Trait Anxiety Inventory I and II (STAI-Y1–STAI-Y2) [19]; Toronto Alexithymia Scale 20 (TAS-20) [20]; Interpersonal Reactivity Index (IRI) [21]; Connor-Davidson Resilience Scale (CD-RISC) [22]; Health Locus of Control scale (HLC) [23].

To evaluate the level of adherence, each participant answered the following question: “Have you ever forgotten to take the medications provided for your treatment plan?” with the dichotomous answer yes/no, in both T0 and T2 phases.

The participants’ renal function at T0, T1, and 3 months after transplant (T2) was evaluated based on creatinine, CDK-EPI (creatinine levels calculated based on age and gender), and azotemia levels, according to the immunosuppressive protocol preferred by the transplant center.

Moreover, during the T0 phase, socio-demographic data and detailed features of the present disease (disease onset, duration and type of dialysis, comorbidities, previous or current psychopathology, schooling, and occupation) for all participants were collected. Afterward, a short interview was conducted related to the medical expenditure in the last year seeking information about the frequency of visits to the general practitioner, hospitalizations and days of hospitalization, admission to the emergency room, drugs stipulated by the therapeutic plan, and consumption of psychotropic drugs, at T0 and T2.

### 2.6. Statistical Analysis

All statistical analyses were carried out using STATISTICA 8.0 StatSoft, Inc. 1994-2007 Tulsa, OK. To evaluate the effectiveness of the expressive writing intervention, a repeated measures (Fisher’s *F*) ANOVA and planned comparisons on the psychological variables, adherence evaluation, renal function indices, and health expenditure between the two randomized groups was carried out. A Chi-square test (χ^2^) was used to test the hypotheses with the dichotomous variable of adherence and with patients’ characteristics.

## 3. Results

Fifty-seven patients were approached for the study, out of which 11 (19%) refused to participate and 11 (19%) were discounted according to the exclusion criteria, resulting in 35 patients being enrolled in the study. The final sample was composed of 26 participants with a mean age of 48.6 ± 14.4 years (Table 1). Three patients received an organ from living donors, while the remaining (*n* = 23) received a cadaveric transplant (Figure 1). All the participants were Caucasian and Italian citizens. The mean and standard deviation values for each analyzed variable are presented in Table 2. Significant scheduled comparisons for psychological and biological variables are shown in Table 3. The two groups, the EWG (*n* = 16) and the CG (*n* = 10), were statistically similar in terms of age distribution (F (1,22) = 0.6, *p* = 0.45) and pre-operative comorbidities, such as diabetes, hypertension and heart disease (Type 1 diabetes: χ^2^ (1) = 0.3, *p* = 0.61 Type 2 diabetes: χ^2^ (1) = 0.3, *p* = 0.61 hypertension: χ^2^ (1) = 0.1; *p* = 0.80 heart disease: χ^2^ (1) = 0.3, *p* = 0.61) (Table 1). The main results of the study are summarized in Figure 2.

### 3.1. Psychological Variables

The effect of the time factor was for BDI-II (T1 < T0), STAI-Y1 (T1 < T0), STAI-Y2 (T1 < T0), IRI-PT (T1 < T0), IRI-PD (T1 < T0), and CD-RISC-control (T2 < T1) (Table 3).

At T1, the planned comparisons showed that the EWG had a significant reduction in depressive symptoms on BDI-II and levels of alexithymia, as seen in TAS-20-F2. Likewise, both groups had significantly lower empathy levels in the “Perspective Thinking” subscale and “Personal Distress” subscale. Interestingly, at T2, only the EWG had increased empathy levels on the “Personal Distress” subscale. Lastly, at T2, there was a significant reduction in the resilience levels, as measured on the CD-RISC-control for both groups.

### 3.2. Adherence Variables

The distribution of the participants who responded to the item “Have you ever forgotten to take the medications provided for your treatment plan in the last 3 months?” for the two groups was as follows: the EWG: (Yes: *n* = 9 at T0, *n* = 4 at T2; No: *n* = 7 at T0, *n* = 12 at T2); CG (Yes: *n* = 7 at T0, *n* = 2 at T2; No: *n* = 3, *n* = 7 at T2) (Table 3). The distribution of the EWG at T2 was significantly different compared to an equal-distribution (Yes: *n* = 4, No: *n* = 12; χ^2^ (1) = 4.0, *p* = 0.045).

### 3.3. Biological Variables

The time factor was significant for CDK-EPI (T1 > T0; T2 > T1), creatinine (T1 < T0; T2 < T1), and azotemia (T2 < T1) (Table 3). A significant effect of group was seen on azotemia levels, with lower levels in the EWG than the CG. This was evident from the planned comparisons, where the EWG at T1 showed a significant decrease in azotemia, creatinine levels, and an increase in the CDK-EPI levels compared to the T0 phase. Similarly, at T1, the CG showed a decrease in creatinine levels and an increase in CDK-EPI levels compared to at T0.

At T1 the EWG showed a higher level of CDK-EPI levels compared to the control group. Furthermore, at T1, the EWG had lower azotemia levels than the CG. In the T2 vs. T1 comparison, the EWG had significantly reduced azotemia levels, and the CG also showed a decrease in azotemia and creatinine levels with increased CDK-EPI levels.

In addition, the EWG and the CG were equally distributed regarding post-operative complications, such as drug toxicity (at T1: χ^2^ (1) = 0.0, *p* = 1.000); at T2: χ^2^ (1) = 1.7, *p* = 0.187) and cytomegalovirus (at T1: χ^2^ (1) = 0.0, *p* = 1.000; at T2: χ^2^ (1) = 1.2, *p* = 0.281).

### 3.4. Health Expenditure

The planned comparisons at T0 showed a significantly greater number of medical visits for the EWG (*p* = 0.030). There were no significant planned comparisons regarding the number of hospitalizations, number of recovery days, and number of access to emergency room.

## 4. Discussion

This study aimed to investigate the psychological and biological effects of expressive writing on renal transplant patients, in order to develop methods that can be used to mitigate the anxiety and depression associated with such major surgical procedures and ensure patient’s adherence, to increase the acceptance of transplanted kidney. 

A main finding of the present study was that the depressive symptoms and alexithymia (in the subscale “describe your emotions”) decreased only in the expressive writing group at T1. Several previous studies have evaluated the effect of a psychological intervention on depressive symptoms in transplanted patients, with a significant decline in depressive symptoms [24], specifically in those who performed an expressive writing treatment [25,26,27]. Regarding alexithymia, the results are debatable, with studies focusing on the role of alexithymia as a moderator of the effects of expressive writing with positive or negative outcomes [28,29], rather than on the effect of expressive writing on alexithymia levels. However, other studies described forms of secondary alexithymia as a response to a psychological distress of an organic disease, rather than a stable personality trait (primary alexithymia) [30]. In the present study, an effect of expressive writing on the ability to describe emotions was observed, which may be explained by the fact that the expressive writing associated with the expectation related to the kidney transplantation may have offered the patients an opportunity to attenuate a secondary alexithymia that may have arisen during the pre-transplantation hemodialysis period [30].

A second finding of the present study was that expressive writing was associated with an increase of adherence to the pharmacological therapy at 3 months after transplantation and with an improvement of renal function, where the CDK-EPI level on the discharge day (T1) in the expressive writing group was significantly higher compared to the control group. Likewise, the azotemia levels also decreased significantly in the patients who performed the expressive writing intervention at T1. Although both groups had significantly reduced creatinine levels at T1, the azotemia and CDK-EPI concentrations suggest a more pronounced improvement of renal function in the patients with the expressive intervention. These results are also supported by the fact that the two groups were quite homogeneous, both for pre-operative comorbidities (diabetes, hypertension and heart disease) and for post-operative complications (cytomegalovirus and drug toxicity). This finding suggests that the better renal function in the expressive writing group was not due to pre-existing medical conditions or post-operative complications. The results of the present study suggest a possible effect of the expressive writing on the depressive symptoms and emotional state in patients undergoing kidney transplantation, confirming the efficacy of expressive writing on the ability to elaborate potential trauma due to critical events in life, such as cancer or related to pregnancy [25,31,32,33,34]. Moreover, the association between the presence of an expressive writing intervention and a higher adherence to the pharmacological treatment with greater kidney functionality suggests the possibility that the expressive writing could also have had an effect, probably mediated by the psychological state and adherence to the immunosuppressive therapy, on the biological acceptance of the organ. 

The association between the presence of an expressive writing intervention and kidney functionality was also suggested by a recent study, which described the effect of expressive writing on the immune system [16]. In a previous study on patients receiving the hepatitis B vaccine, a significant increase in antibodies against hepatitis B was found in patients who underwent a writing expressive intervention compared to the untreated group, as well as an increase in CD-4 type T-lymphocytes the day after the writing task [35]. CD-4 type T-lymphocytes, also known as T-helper cells, seem to have a role in the immune response during the acceptance of a new transplanted organ [36]. Another previous study highlighted the efficacy of a psychological intervention in modifying the body’s inflammatory state, as significantly reduced plasma chemokine concentration in patients with moderate and mild depressive symptoms was found [37]. Chemokines and cytokines act by regulating the function of T lymphocytes, which exert a direct cytotoxic effect through the release of TNF (tumor necrosis factor), which eventually can lead to ischemia in the transplanted tissues and subsequent rejection [38,39]. An intriguing hypothesis to explain these findings could be that expressive writing, by improving psychological state, could indirectly affect the peripheral inflammatory state and consequently increase the probability of the acceptance of the transplanted organ.

The present study also showed that, at T2 (3 months), renal functioning (creatinine and CDK-EPI) remained constant in the EWG and was improved in the CG, while azotemia decreased in both groups. These results indicate that after 3 months from the expressive writing intervention, both groups tended to equivalence in terms of renal functioning. Coherently, a recent meta-analysis that analyzed over 140 studies based on expressive writing found that studies with less than 1-month follow-up had larger psychological health effects than studies with a longer follow-up period (≥1 month) [31]. 

Another factor, empathy, was significantly reduced in both groups on the “perspective thinking” subscale, which assesses the tendency to spontaneously adopt the other’s psychological point of view, and the “personal distress” subscale level, which assesses the tendency to feel distressed in situations when other people are in trouble. However, at T2 (3 months), the empathy levels in the “personal distress” subscale were increased significantly only in the EWG. It is conceivable that in the immediate post-transplant phase, the patient tends to adopt a more self-centered perspective, as an adaptive strategy for receiving and accepting the organ, focusing mainly on the personal moment of difficulty and change, rather than taking a perspective centered on others’ difficulties. Nevertheless, at a later stage, the writing intervention seems to have a significant influence on facilitating the transition from a self-centered perspective to one more concentrated on others. 

An interesting result of the present study concerns the resilience in the “control” subscale (in control of your life), which, in both groups, decreased significantly at T2 (3 months). This result indicates that the fear of potential rejection or death after transplantation could invalidate the perception of having control over one’s life and goals, thus decreasing resilience. A recent review showed how the post-transplant phase is gripped by the fear of rejection, of possible infections, or return to dialysis despite the transplant [40,41]. 

Besides the above-mentioned results, it was seen that the EWG, at T0, turned to their general practitioner more than the CG. This result may be attributed to a chance allotment of the patients who turned to their general practitioner more in the EWG. 

The present study has important limitations. First of all, the analysis was performed on a limited cohort, mainly due to a high rate of drop out. Consequently, the sample size was not sufficient to obtain a relevant effect size, reducing the possibility of generalizing the results to the whole population of patients undergoing kidney transplant. Therefore, it will be necessary to confirm the findings with a greater sample size. Moreover, the measure of adherence used in the present study was very basic. Today, the difficulty in finding instruments adapted to evaluating perceived adherence to a pharmacological treatment in a validated and reliable way remains unsolved [42], confirming the necessity of conducting future studies using more appropriate and validated instruments. The high number of T0 refusals (19%) highlighted the resistance to participating in the study in some patients. Moreover, assigning randomized treatments to the participants did not allow personalization of the psychological treatment.

In conclusion, although the statistical evidence was weak, owing to the small sample size, the findings of the present study suggest the effectiveness of expressive writing on the psychological health, on the adherence to the pharmacological treatment, and on the biological acceptance of the transplanted organ in patients undergoing kidney transplantation. At the same time, this study underlines the need to perform a prolonged or repeated expressive intervention during the three months following kidney transplant. 

The findings of the present study have relevant clinical implications for the possibility of planning a brief psychological intervention aimed at processing emotional involvement and increasing adherence in transplanted patients, as well as a consequent acceptance of the organ and decreased healthcare costs. 

## Figures and Tables

**Figure 1 healthcare-10-01559-f001:**
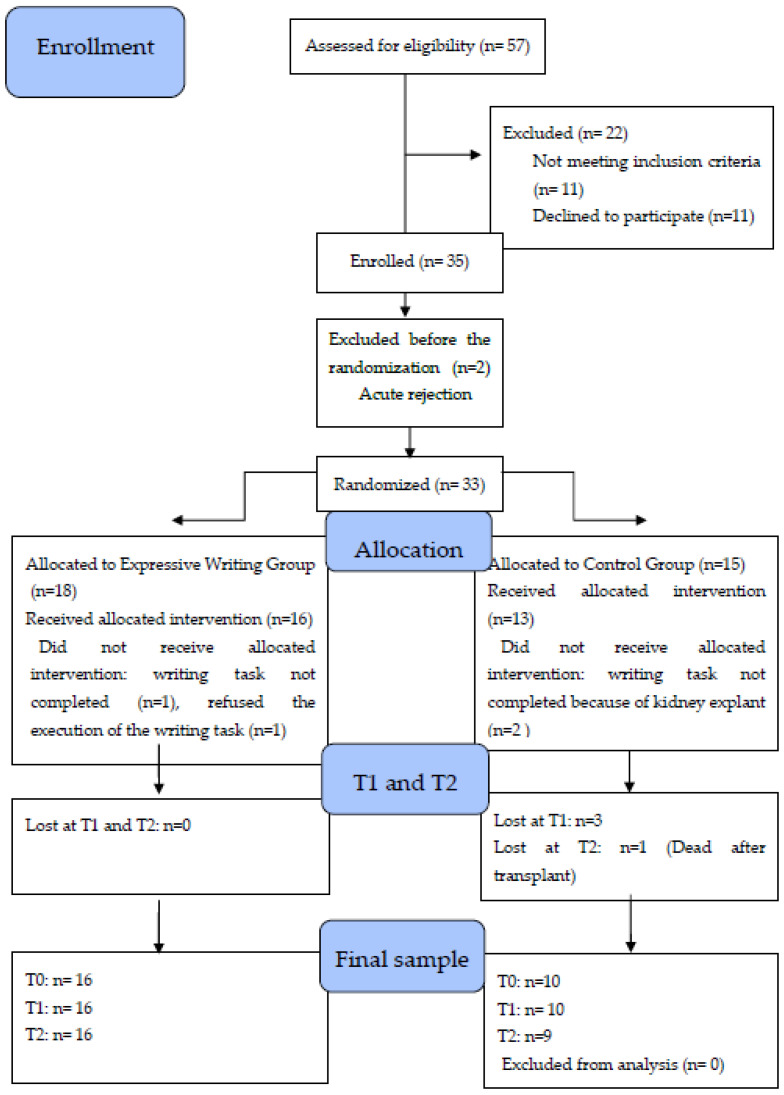
Consort 2010 Flow Diagram of the study.

**Figure 2 healthcare-10-01559-f002:**
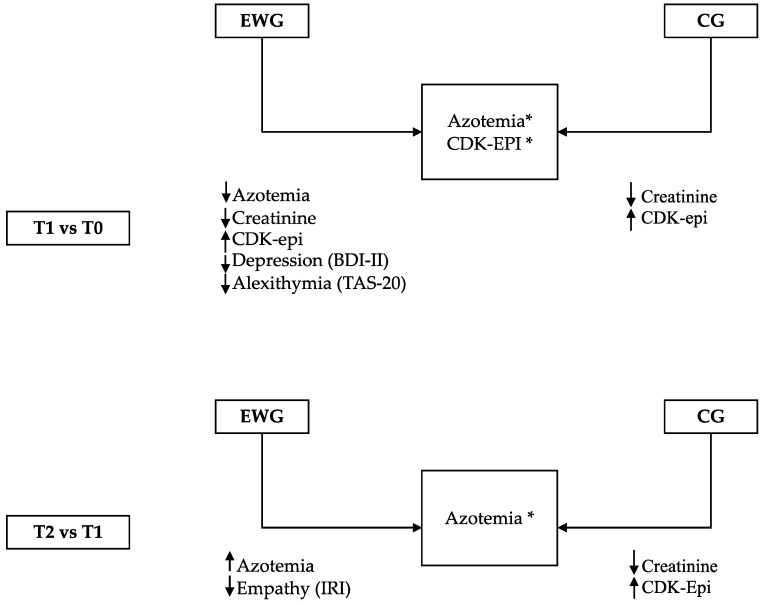
Significant differences between groups (*) and within groups on the investigated variables along the time frame.

**Table 1 healthcare-10-01559-t001:** Characteristics of the sample.

Socio-Demographic and Clinical Parameters	Expressive Writing Group (*n* = 16) *M* ± *SD* or *n* (%)	Control Group (*n* = 10) *M* ± *SD* or *n* (%)	
Age	47.25 ± 16.47	50.80 ± 10.59	*p* = 0.45
Gender (Male vs. Female)	11 vs. 5 (69% vs. 31%)	3 vs. 7 (30% vs. 70%)	*p* = 0.13
Marital Status	Married 4 (25%) Unmarried 8 (50%) Divorced 1 (6%) Widowed 3 (19%)	Married 6 (60%) Unmarried 1 (10%) Divorced 3 (30%) Widowed 0 (0%)	*p* = 0.07
Schooling	Primary school 0 (0%) Secondary school 9 (57%) High School 5 (31%) Degree 2 (12%)	Primary school 1 (10%) Secondary school 3 (30%) High School 5 (50%) Degree 1 (10%)	*p* = 0.07
Employment	Housewife 1 (6%) Unemployed 4 (25%) Employed 7 (44%) Pensioned 4 (25%)	Housewife 3 (30%) Unemployed 2 (20%) Employed 4 (40%) Pensioned 1 (10%)	*p* = 0.38
Psychopathology	Previous 2 (12.5%) Actual 2 (12.5%)	Previous 3 (30%) Actual 1 (10%)	*p* = 0.62
Dialysis	Peritoneal 2 (12.5%) Hemodialysis 12 (75%), of which 1 (8%) had previously received peritoneal dialysis Never received 2 (12.5%)	Peritoneal 1 (10%) Hemodialysis 8 (80%), of which 3 (37%) had previously received peritoneal dialysis Never received 1 (10%)	*p* = 0.95
Comorbidities	Type I Diabetes 1 (6.2%) Type 2 Diabetes 1 (6.2%) Hypertension 4 (25%) Heart disease 1 (6.2%)	Type I Diabetes 0 (0%) Type 2 Diabetes 0 (0%) Hypertension 2 (20%) Heart disease 0 (0%)	*p* = 0.61 *p* = 0.61 *p* = 0.80 *p* = 0.61
Donors’ age	52.12 ± 13.75	55.0 ± 17.57	*p* = 0.64
Operation time	218.12 ± 74.61	213.0 ± 77.21	*p* = 0.87
Intraoperative complications	Absent	Absent	-

**Table 2 healthcare-10-01559-t002:** Averages and standard deviations of psychological variables, adherence, biological variables, and health expenditure variables.

Measures	Expressive Writing Group (EWG)			Control Group (CG)		
	T0 (*n* = 16)	T1 (*n* = 16)	T2 (*n* = 16)	T0 (*n* = 10)	T1 (*n* = 10)	T2 (*n* = 9)
**BDI-II**	5.2 ± 4.7	2.9 ± 3.1	3.8 ± 5.8	4.9 ± 2.33	2.6 ± 2.8	2.8 ± 2.9
**STAI-Y1**	43.8 ± 12.7	37.3 ± 13.5	36.6 ± 13.3	41.3 ± 8.8	36.5 ± 8.5	32.8 ± 7.8
**STAI-Y2**	38.6 ± 9.0	36.1 ± 12.0	32.2 ± 10.8	39.2 ± 8.3	33.7 ± 9.6	28.7 ± 9.0
**TAS-20-F1**	14.7 ± 6.5	16.1 ± 7.3	14.8 ± 8.5	6.4 ± 2.0	13.9 ± 5.2	12.5 ± 5.1
**TAS-20-F2**	15.3 ± 5.7	12.9 ± 4.1	12.4 ± 4.9	11.2 ± 6.3	9.9 ± 4.9	10.1 ± 4.2
**TAS-20-F3**	20.6 ± 4.2	18.8 ± 4.3	17.2 ± 5.1	21.8 ± 4.6	20.2 ± 4.4	19.5 ± 6.2
**TAS-20-TOT**	50.6 ± 13.3	47.8 ± 18.9	45.1 ± 16.9	48.2 ± 11.8	44.0 ± 11.3	42.2 ± 12.6
**IRI-PT**	17 ± 3.4	14.8 ± 3.6	13.9 ± 4.1	18.9 ± 3.5	16.8 ± 3.6	16.7 ± 4.1
**IRI-FS**	14.5 ± 3.6	14.6 ± 4.2	14.2 ± 4.9	16.9 ± 3.5	16 ± 3.8	17.2 ± 2.7
**IRI-EC**	19.5 ± 4.9	19.2 ± 2.7	19.2 ± 4.1	21 ± 4.7	20.4 ± 4.4	22.5 ± 4.3
**IRI-PD**	8.9 ± 4.4	10.6 ± 4.8	12 ± 6.4	9.3 ± 4.71	12.8 ± 4.8	10.4 ± 5.9
**CD-RISC competence**	25.1 ± 4.9	24.6 ± 5.0	24.3 ± 6.1	26.3 ± 4.7	27.8 ± 4.4	26.7 ± 3.8
**CD-RISC emotion managment**	19.9 ± 4.1	19.5 ± 3.5	19.1 ± 5.8	21.1 ± 4.6	21.5 ± 4.3	22.4 ± 3.4
**CD-RISC secure relationship**	13.9 ± 3.7	14.2 ± 2.7	14.6 ± 4.0	15.7 ± 2.5	14.9 ± 2.9	15.5 ± 2.9
**CD-RISC control**	9.1 ± 1.9	9.3 ± 1.8	6.0 ± 1.9	8.8 ± 1.5	8.8 ± 2.0	6.0 ± 1.3
**CD-RISC spiritual**	5.6 ± 1.7	5.6 ± 1.6	5.4 ± 1.6	6.5 ± 1.3	5.9 ± 2.1	6.7 ± 1.9
**HCLI tot**	15.7 ± 5.9	14.6 ± 5.7	14.7 ± 5.0	14.6 ± 4.1	16.3 ± 4.4	16.7 ± 3.2
**HLCE tot**	12.7 ± 5.5	13.2 ± 6.0	16.8 ± 5.6	16.2 ± 6.5	16.9 ± 8.2	16.8 ± 5.6
**Azotemia**	97.2 ± 47.1	79.2 ± 36.4	49.8 ± 10.5	102.7 ± 47.2	110.9 ± 54.6	60.5 ± 19.6
**Creatinine**	7.7 ± 3.2	1.7 ± 0.7	1.5 ± 0.7	7.1 ± 3.6	2.5 ± 1.8	1.4± 0.4
**CDK-EPI**	8.3 ± 3.7	49.7 ± 17.6	56.6 ± 17.2	8.3 ± 3.4	36.8 ± 22.3	53.5 ± 13.3
**Drug toxicity (n^)**	/	0/16	3/16	/	0/10	0/9
**Cytomegalovirus (n^)**	/	0/16	1/16	/	0/10	2/9
**Adherence**	9 yes/7 no	/	4 yes/12 no	7 yes/3 no	/	2 yes/7 no
**General pratictioner visits (n^)**	2.8 ± 3.3	/	0.7 ± 0.4	1.0 ± 1.0	/	0.8 ± 0.4
**Hospitalization (n^)**	0.6 ± 0.5	/	0.5 ± 0.5	0.4 ± 0.5	/	0.4 ± 0.5
**Recovery’s (n^/days)**	11.2 ± 18.3	13.7 ± 9.7	7.7 ± 12.2	12.7 ± 20.6	13.7 ± 10.7	6.7 ± 10.0
**Access to the Emergency Room (n^)**	0.3 ± 0.5	/	0.3 ± 0.5	0.3 ± 0.5	/	0.1 ± 0.3

*Notes*: BDI-II = Beck Depression Inventory II, STAI-Y1 e STAI-Y2 = State Trait Anxiety Inventory 1 (trait) e 2 (state), TAS-20 = Toronto Alexithymia Scale 20 (TAS-20-F1 = Difficulty Identifying Feelings, TAS-20-F2 = Difficulty Describing Feelings, TAS-20-F3 = Externally-Oriented Thinking), IRI = Interpersonal Reactivity Index (IRI-PT = Perspective-Taking IRI-FS = Fantasy Scale, IRI-EC = Empathic Concern, IRI-PD = Personal Distress), CD-RISC = Connor–Davidson Resilience Scale (personal competence, emotion’s management, secure relationships, Control, Spirituality), HLC = Health Locus of control scales (HCLE = External Locus of Control, HCLI= Internal Locus of Control), CDK-EPI = creatinine levels calculated based on age and gender.

**Table 3 healthcare-10-01559-t003:** Repeated Measure ANOVA Expressive Writing Group vs. Control Group *per* 3 times (T1 vs. T0 and T2 vs. T1) on significant psychological variables (BDI-II, STAI-Y1, STAI-Y2, TAS-20, IRI, CD-RISC) and on significant biological variables (Azotemia, Creatinine, CDK-EPI). In (**a**) are reported between group and within group planned comparisons related to the two at T1 vs. T0 time; in (**b**) are reported between group and within group planned comparisons related to the two at T2 vs. T1 time.

	(a)		
	T1 vs. T0 (Expressive Writing Group (EWG) N = 16 vs. Control Group (CG) N = 10)		
	Effects	Planned Comparison Between Group	Planned Comparison Within Group
**Azotemia**	F Group (1, 24) = 1.13; *p* = 0.298 F Time (1, 24) = 0.60; *p* = 0.446 F Int GxT (1, 24) = 4.00; *p* = 0.057	EWGT0 < CGT0 *p* = 0.798 EWGT1 < CGT1 *p* = 0.095	EWGT0 > EWGT1 *p* = 0.035 CGT0 < GCT1 *p* = 0.442
**Creatinine**	F Group (1, 24) = 0.02; *p* = 0.876 F Time (1, 24) = 65.09; *p* = 0.000000 F Int GxT (1, 24) = 1.25; *p* = 0.274	EWGT0 > CGT0 *p* = 0.555 EWGT1 < CGT1 *p* = 0.407	EWGT0 > EWGT1 *p* < 0.001 CGT0 > CGT1 *p* = 0.00017
**CDK-EPI**	F Group (1, 24) = 2.58; *p* = 0.121 F Time (1, 24) = 77.08; *p* = 0.000000 F Int GxT (1, 24) = 2.61; *p* = 0.119	EWGT0 > CGT0 *p* = 0.995 EWGT1 > CGT1 *p* = 0.027	EWGT0 < EWGT1 *p* < 0.001 CGT0 < CGT1 *p* = 0.027
**BDI II**	F Group (1, 24) = 0.08; *p* = 0.771 F Time (1, 24) = 7.71; *p* = 0.010 F Int GxT (1, 24) = 0.00006; *p* = 0.994	EWGT0 > CGT0 *p* = 0.808 EWGT1 > CGT1 *p* = 0.815	EWGT0 > EWGT1 *p* = 0.034 CGT0 > CGT1 *p* = 0.090
**STAI Y1**	F Group (1, 24) = 0.109; *p* = 0.743 F Time (1, 24) = 5.29; *p* = 0.030 F Int GxT (1, 24) = 0.05; *p* = 0.818	EWGT0 > CGT0 *p* = 0.690 EWGT1 > CGT1 *p* = 0.863	EWGT0 > EWGT1 *p* = 0.523 CGT0 > CGT1 *p* = 0.199
**STAI Y2**	F Group (1, 24) = 0.06; *p* = 0.800 F Time (1, 24) = 4.81; *p* = 0.038 F Int GxT (1, 24) = 0.67; *p* = 0.419	EWGT0 < CGT0 *p* = 0.888 EWGT1 > CGT1 *p* = 0.553	EWGT0 > EWGT1 *p* = 0.280 CGT0 > CGT1 *p* = 0.067
**TAS-20 F2**	F Group (1, 24) = 3.56; *p* = 0.071 F Time (1, 24) = 4.02; *p* = 0.056 F Int GxT (1, 24) = 0.344; *p* = 0.562	EWGT0 > CGT0 *p* = 0.588 EWGT1 > CGT1 *p* = 0.158	EWGT0 > EWGT1 *p* = 0.047 CGT0 > CGT1 *p* = 0.374
**IRI PT**	F Group (1, 24) = 2.39; *p* = 0.135 F Time (1, 24) = 11.63; *p* = 0.002 F Int GxT (1, 24) = 0.05; *p* = 0.945	EWGT0 < CGT0 *p* = 0.184 EWGT1 < CGT1 *p* = 0.165	EWGT0 > EWGT1 *p* = 0.0097 CGT0 > CGT1 *p* = 0.043
**IRI PD**	F Group (1, 24) = 0.52; *p* = 0.477 F Time (1, 24) = 25.20; *p* = 0.00004 F Int GxT (1, 24) = 2.80; *p* = 0.107	EWGT0 < CGT0 *p* = 0.255 EWGT1 < CGT1 *p* = 0.822	EWGT0 > EWGT1 *p* = 0.012 CGT0 > CGT1 *p* = 0.00003
	**(b)**		
	**T2 vs. T1** **(Expressive Writing Group (EWG) N = 16 vs. Control Group (CG) N = 9)**		
	**Effects**	**Planned Comparison** **Between Group**	**Planned Comparison** **Within Group**
**Azotemia**	F Group (1, 23) = 4.60; *p* = 0.043 F Time (1, 23) = 18.05; *p* = 0.0003 F Int GxT (1, 23) = 1.33; *p* = 0.260	EWGT1 < CGT1 *p* = 0.023 EWGT2 < CGT2 *p* = 0.443	EWGT1 > EWGT2 *p* = 0.017 CGT1 > CGT2 *p* = 0.003
**Creatinine**	F Group (1, 23) = 0.641; *p* = 0.431 F Time (1, 23) = 6.14; *p* = 0.021 F Int GxT (1, 23) = 3.20; *p* = 0.087	EWGT1 < CGT1 *p* = 0.101 EWGT2 > CGT2 *p* = 0.070	EWGT1 < EWGT2 *p* = 0.572 CGT1 > CGT2 *p* = 0.014
**CDK-EPI**	F Group (1, 23) = 1.10; *p* = 0.304 F Time (1, 23) = 7.19; *p* = 0.013 F Int GxT (1, 23) = 0.803; *p* = 0.379	EWGT1 > CGT1 *p* = 0.171 EWGT2 > CGT2 *p* = 0.677	EWGT1 < EWGT2 *p* = 0.150 CGT1 < CGT2 *p* = 0.035
**IRI PD**	F Group (1, 23) = 0.28; *p* = 0.603 F Time (1, 23) = 3.36; *p* = 0.079 F Int GxT (1, 23) = 2.96; *p* = 0.098	EWGT1 < CGT1 *p* = 0.851 EWGT2 > CGT2 *p* = 0.255	EWGT1 < EWGT2 *p* = 0.0085 CGT1 < CGT2 *p* = 0.942
**CDRISC control**	F Group (1, 23) = 0.77; *p* = 0.388 F Time (1, 23) = 48.28; *p* = 0.000000 F Int GxT (1, 23) = 0.008; *p* = 0.928	EWGT1 > CGT1 *p* = 0.523 EWGT2 > CGT2 *p* = 0.455	EWGT1 > EWGT2 *p* < 0.001 CGT1 > CGT2 *p* = 0.00015

*Notes*: F-Group = Group Effect (Expressive Writing Group vs. Control Group), F-Time= Time Effect (T0 vs. T1 vs. T2), F-Int GxT= Interaction Group per Time Effect. BDI-II = Beck Depression Inventory II, STAI-Y1 e STAI-Y2 = State Trait Anxiety Inventory 1 (trait) e 2 (state), TAS-20 = Toronto Alexithymia Scale 20 (TAS-20-F2 = Difficulty Describing Feelings), IRI = Interpersonal Reactivity Index (IRI-PT = Perspective-Taking IRI-PD = Personal Distress), CD-RISC = Connor–Davidson Resilience Scale (Control), CDK-EPI = creatinine levels calculated based on age and gender.

## Data Availability

Data are available on request, due to privacy and ethical restrictions.

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
