# Peer review of "Effectiveness of Expressive Writing in Kidney Transplanted Patients: A Randomized Controlled Trial Study"

_healthcare, 2022, doi:10.3390/healthcare10081559_

Round 1

Reviewer 1 Report

In this article, the authors assessed the effectiveness of the expressive writing intervention on the psycological and  physiological variables after kidney transplant. Outcomes of the study were depression, anxiety, alexithymia, empathy, resilience, locus of control, creatinine, CDK-EPI, and azotemia. The main problems of this article derives from the observation of the content of Tab.1. The patients’s characteristics are hetereogeneous between the two groups. In fact, age, gender, marital status, education, employment, dialysis and comorbidities materially different between EW group and  C group. Thus, the random allocation of partecipants is questionable. Futhermore, by definition, a clinical trial is useful for comparing the differences in the effect of an "intervention" between two groups (treated / untreated). Unfortunately, in this article, some statistical test are performend within the same group but not between the two groups. In my opinion, the study must improved by increasing the sample size, randomizing the two groups better, adapting the statistical analysis to the study design and perhaps focusing on a smaller number of variables.

Author Response

Reviewer 1:

In this article, the authors assessed the effectiveness of the expressive writing intervention on the psycological and  physiological variables after kidney transplant. Outcomes of the study were depression, anxiety, alexithymia, empathy, resilience, locus of control, creatinine, CDK-EPI, and azotemia. The main problems of this article derives from the observation of the content of Tab.1. The patients’s characteristics are hetereogeneous between the two groups. In fact, age, gender, marital status, education, employment, dialysis and comorbidities materially different between EW group and  C group. Thus, the random allocation of partecipants is questionable.

Authors:

The authors thank the reviewer for her/his suggestion. Following her/his comments the authors added in Table 1 the p-values for each patients’ characteristics.

Reviewer 1:

Futhermore, by definition, a clinical trial is useful for comparing the differences in the effect of an "intervention" between two groups (treated / untreated). Unfortunately, in this article, some statistical test are performend within the same group but not between the two groups.

Authors:

The authors thank the reviewer for her/his suggestion. Following her/his comments the authors replaced Table 3 with two tables: Table 3a (T1 vs T0) and Table 3b (T2 vs T1) with two different columns, one for “within group planned comparisons” and one for “between group planned comparisons”.

Reviewer 1:

In my opinion, the study must improved by increasing the sample size, randomizing the two groups better, adapting the statistical analysis to the study design and perhaps focusing on a smaller number of variables.

Authors:

Thank you for the reviewer’s opinion. Unfortunately at today it is not possible to re-open the study because the randomization list was completed. Moreover, the number of foreseen participants was based on a-priori power analysis. According to the reviewers’ opinion this was considered as the main limitation of the study as reported in the discussion section.

Reviewer 2 Report

Authors performed a randomized control trial to evaluate the effect of expressive writing on psychological variables and on the biological markers of renal function after transplant. They found positive effects of EWG on some of the clinical indicators. This is an interesting clinical topic and might be a helpful tool to implement in patients with kidney transplant. The weakness of the study is being very less number of patients but given this a RCT, should give merit to the findings. I don't have any suggestions to change in the manuscript. 

Author Response

Reviewer 2:

Authors performed a randomized control trial to evaluate the effect of expressive writing on psychological variables and on the biological markers of renal function after transplant. They found positive effects of EWG on some of the clinical indicators. This is an interesting clinical topic and might be a helpful tool to implement in patients with kidney transplant. The weakness of the study is being very less number of patients but given this a RCT, should give merit to the findings. I don't have any suggestions to change in the manuscript. 

Authors:

the authors really appreciate the reviewer’s time spent revising this article and for the positive comments reported.

Reviewer 3 Report

I am honored to review this fine manuscript from the Italian Group. This is a randomized control trial showing the effectiveness of Expressive Writing (EW) intervention for recipients who received kidney transplantation. The authors stated that depressive symptoms and alexithymia levels after transplantation decreased by intervention of EW, with better adherence.  They also stated that EW intervention was significantly associated with higher CDK-EPI indicating better renal functioning as well as  improving the psychological health of kidney transplant recipients.  Their report might have important information for all transplant surgeons and recipients with severe renal diseases. However, I have some major concerns about this manuscript.

Major comments

·      My biggest concern is that all the tables are very confusing and not well organized. The author should use simpler tables that clearly show the points that the authors want to emphasize.  In addition, the use of valid figures may be a better way to help readers effectively understand the prominent findings of this study. I strongly recommend that the tables be modified and that new figures be created.

·      In the study design, of the 57 candidates, only 33 (58% of all) were enrolled in the study.  This study excluded recipients from the cohort who had severe psychopathological conditions or cognitive impairments, and 9 others were dropped from the cohort.  Do the authors believe that the study design with nearly half of the candidates dropping out of research is appropriate? Furthermore, do the authors think that the only 33 cases analyzed in this study represent all kidney transplant patients? I believe that this study should be conducted with a larger number of cases if possible. If not, the authors should state clearly that this study was analyzed on a limited cohort in the limitations.

·      The authors found that EW intervention was associated with an improvement in renal function, with the CDK-EPI level on the discharge day. I believe that EW interventions for kidney transplant recipients could be effective in adhering to their medications and improving renal function. However, the authors should also clearly state the differences between other factors in the control group and the EW group. I could not find surgical information (blood lost in operation, operation time, intraoperative complications, etc.) or donor information (donor characteristics, quality of grafts) in this manuscript. I think that these factors strongly affect post-transplant renal function, and the authors need to clearly show any differences between the two groups (EW group vs. Control group).

·      The authors stated that these results are also supported by the fact that the two groups were quite homogeneous  both for pre-operative comorbidities (diabetes, hypertension and heart disease) and for post-operative complications (cytomegalovirus and drug toxicity) in Line 253. However, the authors did not show that there was no statistical difference between these two groups. In the characteristics of recipients in Table 1, there seem to be some differences between the two groups in terms of gender and marriage. My question is whether we can find any statistically significant differences between these two groups? I think it's better to put the p value between the two groups in Table 1.

Minor comments

·      Figure 1. “ Assessed for eligibility (n=”  n is missing.

·      Table 3. Needs annotations for F-Group, F-Time and F-Int GxT.

Author Response

Reviewer 3:

I am honored to review this fine manuscript from the Italian Group. This is a randomized control trial showing the effectiveness of Expressive Writing (EW) intervention for recipients who received kidney transplantation. The authors stated that depressive symptoms and alexithymia levels after transplantation decreased by intervention of EW, with better adherence.  They also stated that EW intervention was significantly associated with higher CDK-EPI indicating better renal functioning as well as  improving the psychological health of kidney transplant recipients.  Their report might have important information for all transplant surgeons and recipients with severe renal diseases. However, I have some major concerns about this manuscript.

Authors:

The authors would like to thank the reviewer for the thorough review that allowed to improve their manuscript.

Reviewer 3:

Major comments

My biggest concern is that all the tables are very confusing and not well organized. The author should use simpler tables that clearly show the points that the authors want to emphasize.  In addition, the use of valid figures may be a better way to help readers effectively understand the prominent findings of this study. I strongly recommend that the tables be modified and that new figures be created.

Authors:

Following the reviewer’s comment the authors replaced Table 3 with two tables: Table 3a (T1 vs T0) and Table 3b (T2 vs T1) with two different columns, one for “within group planned comparisons” and one for “between group planned comparisons”. Moreover, a new figure has been inserted in order to highlight the main results of the study (see figure 2).

Reviewer 3:

In the study design, of the 57 candidates, only 33 (58% of all) were enrolled in the study.  This study excluded recipients from the cohort who had severe psychopathological conditions or cognitive impairments, and 9 others were dropped from the cohort.  Do the authors believe that the study design with nearly half of the candidates dropping out of research is appropriate? Furthermore, do the authors think that the only 33 cases analyzed in this study represent all kidney transplant patients? I believe that this study should be conducted with a larger number of cases if possible. If not, the authors should state clearly that this study was analyzed on a limited cohort in the limitations.

Authors:

Following the reviewer’s suggestion the authors added the following sentence in the discussion section “The present study has important limitations. First of all, the analysis was performed on a limited cohort mainly due to a high rate of drop out. Consequently, the sample size was not sufficient to obtain relevant effect size reducing the possibility to generalize the results on the whole population of patients undergoing kidney transplant.”.

Reviewer 3:

The authors found that EW intervention was associated with an improvement in renal function, with the CDK-EPI level on the discharge day. I believe that EW interventions for kidney transplant recipients could be effective in adhering to their medications and improving renal function. However, the authors should also clearly state the differences between other factors in the control group and the EW group. I could not find surgical information (blood lost in operation, operation time, intraoperative complications, etc.) or donor information (donor characteristics, quality of grafts) in this manuscript. I think that these factors strongly affect post-transplant renal function, and the authors need to clearly show any differences between the two groups (EW group vs. Control group).

Authors:

The authors thank the reviewer for her/his suggestion. Following her/his comments, surgical information (operation time, intraoperative complications) and donor age have been added in Table 1. Moreover, p-values for the comparison between groups have been added.

Reviewer 3:

The authors stated that these results are also supported by the fact that the two groups were quite homogeneous  both for pre-operative comorbidities (diabetes, hypertension and heart disease) and for post-operative complications (cytomegalovirus and drug toxicity) in Line 253. However, the authors did not show that there was no statistical difference between these two groups. In the characteristics of recipients in Table 1, there seem to be some differences between the two groups in terms of gender and marriage. My question is whether we can find any statistically significant differences between these two groups? I think it's better to put the p value between the two groups in Table 1.

Authors:

The authors thank the reviewer for her/his suggestion. Following her/his comments the authors added in Table 1 the p values for pre-operative characteristics.

Reviewer 3:

Minor comments

Figure 1. “ Assessed for eligibility (n=”  n is missing.

Authors:

The missing point “n=57” has been added.

Reviewer 3:

Table 3. Needs annotations for F-Group, F-Time and F-Int GxT.

Authors:

The annotations have been added.

Round 2

Reviewer 1 Report

The authors revised the article by improving the statistical models. However, as the authors also confirm, the patient sample is too small for a clinical study. I suggest you specify this important limitation in your conclusions. Good work.

Author Response

Reviewer 1

Comments and Suggestions for Authors

The authors revised the article by improving the statistical models. However, as the authors also confirm, the patient sample is too small for a clinical study. I suggest you specify this important limitation in your conclusions. Good work.

Authors: The authors thank the reviewer for the comment. Following the suggestion, a sentence related to the sample size has been added in the conclusion.

Reviewer 3 Report

I have carefully read the revised manuscript and the authors’ comments. The authors have appropriately responded to my comments and also indicated their response in the revised manuscript.

I am very grateful for their understanding and for making these changes.  I think most changes are appropriate. However, they should consider making Table2, 3a, and 3b simpler. I think the authors need to show the reader more concise tables focusing on the important parameters rather than showing all the parameters. 

Author Response

Reviewer 3

Comments and Suggestions for Authors

I have carefully read the revised manuscript and the authors’ comments. The authors have appropriately responded to my comments and also indicated their response in the revised manuscript.

I am very grateful for their understanding and for making these changes.  I think most changes are appropriate. However, they should consider making Table2, 3a, and 3b simpler. I think the authors need to show the reader more concise tables focusing on the important parameters rather than showing all the parameters. 

Authors: the authors thank the reviewer for the precious suggestion. Following the reviewers’ comment the Table 2 has been reduced by eliminating the column of total means and standard deviations. Then, Table 3a,3b have been modified focusing only on significant variables.
